

# Dauer fate in a *Caenorhabditis elegans* Boolean network model

Alekhya Kandoor[1] and Janna Fierst[2]

[1] Biomedical Engineering, University of Virginia, Charlottesville, VA, United States of America
[2] Biomolecular Sciences Institute and Department of Biology, Florida International University, Miami, FL, United States of America

## ABSTRACT

Cellular fates are determined by genes interacting across large, complex biological networks. A critical question is how to identify causal relationships spanning distinct signaling pathways and underlying organismal phenotypes. Here, we address this question by constructing a Boolean model of a well-studied developmental network and analyzing information flows through the system. Depending on environmental signals *Caenorhabditis elegans* develop normally to sexual maturity or enter a reproductively delayed, developmentally quiescent 'dauer' state, progressing to maturity when the environment changes. The developmental network that starts with environmental signal and ends in the dauer/no dauer fate involves genes across 4 signaling pathways including cyclic GMP, Insulin/IGF-1, TGF-$\beta$ and steroid hormone synthesis. We identified three stable motifs leading to normal development, each composed of genes interacting across the Insulin/IGF-1, TGF-$\beta$ and steroid hormone synthesis pathways. Three genes known to influence dauer fate, *daf-2*, *daf-7* and *hsf-1*, acted as driver nodes in the system. Using causal logic analysis, we identified a five gene cyclic subgraph integrating the information flow from environmental signal to dauer fate. Perturbation analysis showed that a multifactorial insulin profile determined the stable motifs the system entered and interacted with *daf-12* as the switchpoint driving the dauer/no dauer fate. Our results show that complex organismal systems can be distilled into abstract representations that permit full characterization of the causal relationships driving developmental fates. Analyzing organismal systems from this perspective of logic and function has important implications for studies examining the evolution and conservation of signaling pathways.

## INTRODUCTION

Cellular states, developmental pathways and organismal phenotypes are determined by interactions between biomolecules. These biological networks are large and complex, involving genes, proteins and regulatory molecules spanning diverse pathways and levels of biological organization (*Weng, Bhalla & Iyengar, 1999*). Data-based network inference often implicates large numbers of biomolecules through correlation-based methods (*Stolovitzky, Monroe & Califano, 2007*) but network modeling and analysis

Corresponding author
Janna Fierst, jfierst@fiu.edu

indicate causal relationships are the key to controlling information flow and determining cellular fates (*Zanudo & Albert, 2015*). A fundamental question is how to identify the critical, causal relationships within complex biological systems and between distant nodes in a network (*Lawyer, 2015*). Here, we construct a Boolean model of a well-studied model system and use network analysis techniques to identify the core interactions responsible for the phenotype.

*C. elegans* is a small (∼1 mm), free-living nematode worm that has been used as a model organism in biological research for over 50 years (*Girard et al., 2007*). Populations are androdioecious, composed of self-fertile hermaphrodites and a small number of males, resulting in straightforward laboratory culture and rapid propagation. *C. elegans* are transparent, facilitating microscope study, and eutelic, meaning that all adults have the same number of somatic cells. In *C. elegans* hermaphrodites there are 959 somatic cells while males have 1,031 (*Corsi, Wightman & Chalfie, 2015*). The entire life cycle can be completed within 3 days and these factors have led to the establishment of *C. elegans* as an important model organism for studying development, genetics and essential biological processes. *C. elegans* was the first metazoan to have its genome fully sequenced (*Consortium, 1998*; *Waterston, Sulston & Coulson, 1997*) and the long history of *C. elegans* research has provided a wealth of genetic, genomic and developmental understanding of this small worm (*Riddle et al., 1997*; *Kemphues & Strome, 1997*; *Schedl, 1997*; *Schnabel & Priess, 1997*).

One of the most studied developmental processes in *C. elegans* is dauer (*Hu, 2005-2018*; *Cassada & Russell, 1975*). The small worms are vulnerable to environmental conditions including lack of food or moisture (*Golden & Riddle, 1984a*) and high temperature (*Ailion & Thomas, 2000*). *C. elegans* senses its environment through chemoreceptors that perceive biomolecules including pheromones produced by other worms (*Jeong et al., 2005*). In low-stress environments the worms progress through discrete larval stages L1–L4, to adulthood and reproductive maturity (*Hu, 2005-2018*). Environmental triggers can activate a developmental switch at the L1 stage and lead to dauer arrest (*Vowels & Thomas, 1992*). Dauer worms can move but are developmentally quiescent and survive without feeding for periods of up to 4 months.

*C. elegans* can be forced into the dauer fate with high pheromone concentrations, high temperature or starvation in the lab but the signals can also interact quantitatively in the worm's natural environment (*Jeong et al., 2005*). The dauer pheromone itself is a complex mix of ascarosides (*Butcher et al., 2007*; *Butcher et al., 2008*) with natural variation in production, composition and response between populations (*Viney & Harvey, 2017*), and species (*Mayer & Sommer, 2011*). *C. elegans* strains have substantial, quantitative natural variation for dauer entry (*Harvey, Shorto & Viney, 2008*; *Viney, Gardner & Jackson, 2003*) and exit (*Bubrig, Sutton & Fierst, 2020*) but for the individual worm dauer is an irreversible developmental decision. Undergoing dauer has lasting effects on worm reproduction (*Ow & Hall, 2020*), physiology (*Ow, Nichitean & Hall, 2021*) and locomotion (*Pradhan et al., 2019*).

Dauer is a unique, discrete developmental switch and the molecular basis has been studied extensively. Over 100 genes across four signaling pathways (cyclic GMP, Insulin/IGF-1, TGF-$\beta$ and steroid hormone synthesis) have been implicated as influencing
dauer (*Hu, 2005-2018*; *Gilabert et al., 2016*). Despite extensive descriptive characterization, we lack a robust understanding of the interplay between these pathways and the causal links that drive the system to one fate or another. Here, we study a Boolean model and identify the critical interactions determining dauer fate.

Simplified representations of complex biological networks can accurately distill systems down to essential interactions (*Bornholdt, 2005*). Boolean networks are graphs where nodes represent genes and edges are regulatory interactions defined as Boolean functions or logic gates, for example the operators 'AND' or 'OR' (*Bornholdt, 2008*). Boolean models of biological processes represent states as discrete 'on' or 'off' (*Wang, Saadatpour & Albert, 2012*) and accurately represent the switch-like nature of dauer induction. Regulatory interactions are threshold-gated, qualitative relationships. Here, we use our Boolean network to emulate cellular behavior observed in experimental settings and systematically perturb the system to discover causal links within the larger complex network. We identify the critical interactions and combinations of gene states that determine dauer fate. Evolutionary searches of pathway conservation and divergence typically examine gene-by-gene, sequence level homology (*Pires-daSilva & Sommer, 2003*). Here, we suggest that causal logic and function may be the true phenotypes underlying the evolution of signaling pathways.

## METHODS

### Constructing the dauer network

We first extracted 85 dauer-involved genes described in the WormBook 'Dauer' chapter (*Hu, 2005-2018*). For each of these genes, we then searched on WormBase (*Harris et al., 2019*) for genes known to interact with these dauer genes and genes known to act in the same pathways as these dauer genes. We focused on regulatory interactions and included those for which we identified information regarding the direction of the interaction, for example reported relationships between effector proteins and targets. We did not include genetic interactions or genetic masking/enhancing/suppressing effects between genes. Some of our gene pairs may have reported genetic interactions along with regulatory interactions but we focused on regulatory interactions for constructing our model.

We next surveyed the literature for regulatory interactions within this gene and pathway set and created a filtered set of dauer genes and main-effect regulatory interactions (Table S1). For example, of the 40 insulin signaling genes in the *C. elegans* genome few genes have been reported to act independently in the Insulin/IGF-1 pathway (*Girard et al., 2007*). We condensed the larger dauer-involved set down to 26 genes, each part of the four dauer-inducing pathways represented as nodes in our Boolean model, and connected by 34 edges. We included the nodes 'pheromone' (*Golden & Riddle, 1984b*) and '*cmk-1*' (a food receptor *Neal et al., 2015*) as inputs to the system. These interactions were mainly obtained from the WormBase database (*Harris et al., 2019*) and verified with the scientific publications cited on WormBase (a full list of nodes, interactions, and citations is given in Table S1).

We used this regulatory interaction information for each gene to formulate Boolean functions (Table S2). We combined literature-reported regulations into a preliminary

model and used results obtained from simulations and perturbations to fine-tune the Boolean functions for each node until we were able to mimic the expected behavior of the system. For example, multiple articles reported that *daf-16* was activated either when *akt* was repressed (*Henderson & Johnson, 2001*; *Lin et al., 2001*; *Lee, Hench & Ruvkun, 2001*; *Paradis & Ruvkun, 1998*) or when *daf-12* was activated (*Matyash et al., 2004*; *Dowell et al., 2003*). Simulated conditions for normal adult development should therefore result in *daf-16* repression and no dauer activation. To achieve this known output for *daf-16* we assigned an 'OR' operator for these two regulators. In some cases, we were able to extract the relationship between regulatory nodes from the literature. This approach has been similarly demonstrated in previous works to gain qualitative insights using discrete networks (*Hopfensitz et al., 2013*; *Bornholdt, 2008*).

## Studying system trajectories

The initial conditions used to generate the trajectories of the system's dynamics were based on well-known environmental cues such as food availability (*cmk-1*) and dauer pheromone (pher) detected by *C. elegans* (*Jeong et al., 2005*; *Neal et al., 2015*). Favorable conditions allowed the organism to develop into an adult ("no dauer") and unfavorable conditions into "dauer" (*Golden & Riddle, 1984b*). We set the initial conditions as *cmk-1* = 1 and pher = 0 for normal development (here, dauer = 0) and *cmk-1* = 0 and pher = 1 for dauer fate (here, dauer = 1). We monitored the dynamics and convergence of the system based on the state of the 'dauer' output node (trajectories for each simulation are shown in Fig. S2). Temperature is another well-known regulator that induces dauer formation. We chose to focus here on dauer as a response to resource availability and our model does not include temperature as an input.

We performed simulations with asynchronous updating using boolean2 software (*Albert et al., 2008*). In order to explore system dynamics, the Boolean functions for each gene needs to be updated at each timestep to trace out the trajectory of the system. These transitions for the Boolean functions were carried out using an asynchronous scheme that randomly assigned the states of the nodes. Biological events do not occur at the same time and the timescale at which the events occur can be diverse. Here, we represented this with an asynchronous update scenario. In this framework a collector class averaged the states of nodes over multiple runs for simulating a system with a non-deterministic updating scheme. We performed 100 simulations for each set of initial conditions and for every iteration, the collector averaged the state of each node generated for 1,000 steps. In all our simulations, the nodes reached equilibrium after approximately 15 iterations (trajectories are shown in Fig. S2). Nodes with states > 0.5 were considered 'ON' (as indicated by *Albert et al., 2008*) and nodes with states < 0.5 were considered 'OFF'.

## Validating the model

To validate the model, we checked whether the model has the ability to reproduce known experimental outcomes. Mutant phenotypes in *C. elegans* that lead to either adult or dauer formation are well studied (*Hu, 2005-2018*). We used published experimental evidence to check whether our model could reproduce similar results. For example,

dauer-constitutive (Daf-c) mutants lead to dauer formation in conditions that do not favor dauer while dauer-defective (Daf-d) mutants will not develop into dauer larvae (*Vowels & Thomas, 1992*; *Vowels & Thomas, 1994*; *Gottlieb & Ruvkun, 1994*; *Albert & Riddle, 1988*). We knocked down known Daf-c and Daf-d genes to validate that our dauer Boolean model captured essential wild type and mutant phenotypes (Fig. S1B).

Our model simulations helped us identify the attractor space and stable motifs. A Boolean model will eventually settle down into a static state where all the nodes are either 'on' or 'off' and remain in that state through further iterations. This state of the model is called an attractor and in biological modeling attractors depict specific biological phenotypes. Within this, a set of nodes within the network forms cycles which stay in specific states, and these are called stable motifs. We can represent a network as a reduced system using the nodes that are critical for controlling the stable motifs. We can then study the model dynamics using the reduced form and can make predictions using simpler representations of the model.

Attractors of Boolean networks represent biological phenotypes, here the 'dauer' and 'no dauer' fates. We explored the set of attractors and stable motifs for our model by iterating through all possible combinations of initial conditions and perturbations using pystablemotifs software (*Rozum et al., 2021b*; *Rozum et al., 2021a*). A perturbation of a node in our experiments refers to reversing the existing state of a node and specify that along with the initial conditions set. The network diagrams were created using Cytoscape (*Shannon et al., 2003*). Code, simulations and scripts are available at https://github.com/alekhyaa2/Dauer-Boolean-Model.

## RESULTS

### Constructing a dauer Boolean network

Our exhaustive literature survey identified essential regulatory interactions among dauer pathway genes. From these we constructed a Boolean network with 26 nodes and 34 interactions (Fig. 1A; Table S1). The nodes 'pheromone' (*Golden & Riddle, 1984b*) and '*cmk-1*' (a food receptor (*Neal et al., 2015*)) were input signals that allowed the model to converge onto the dauer/no dauer output node.

### Validating the model by replicating Daf-c and Daf-d phenotypes

In *C. elegans* mutant strains can be categorized as dauer-constitutive (Daf-c), genotypes which produce dauer arrest even in the absence of pheromone and presence of food, and Dauer-defective (Daf-d), genotypes which do not enter dauer arrest under typical dauer-favorable conditions (*i.e.,* pheromone present and/or food limited) (*Vowels & Thomas, 1992*; *Vowels & Thomas, 1994*; *Gottlieb & Ruvkun, 1994*; *Albert & Riddle, 1988*). We validated our model by knocking down known Daf-c genes (*daf-2*, *daf-7*, *daf-11*, and *daf-9*) separately under initial conditions (*cmk-1* = 1 and pher = 0) that do not typically favor dauer and found that these mutants followed the Daf-c phenotype and entered dauer arrest. We also knocked down known Daf-d mutant genes (*daf-12* and *daf-16*) under conditions that typically favor dauer (here, *cmk-1* = 0 and pher = 1) and found these replicated the Daf-d phenotype, failing to enter dauer arrest.

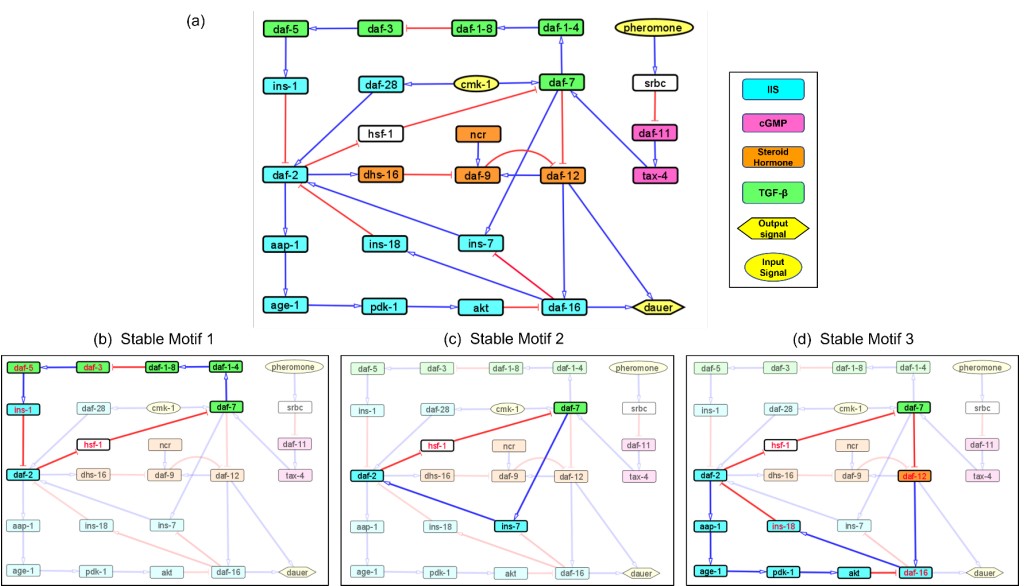

**Figure 1** **Models representing the four major dauer inducing pathways and the stable motifs of this network.** (A) The dauer Boolean network included 26 nodes and 34 edges. Each node represented a molecular entity (gene, receptor, protein, *etc.*) involved in signal transduction pathways. Edges indicated regulatory interactions between the nodes that activated (signified by blue arrow heads) or inhibited (signified by red 'T' edges). The four dauer inducing pathways were represented by four different colors. Here, *cmk-1* and pheromone were input signals which when turned on and off, decided the dauer fate. Nodes *srbc* (pheromone receptor) and *hsf-1* (heat shock protein) did not belong to any of the four dauer pathways. (B, C and D) Subgraphs that were part of the cyclic SMs 1, 2 and 3 respectively, into which the dauer Boolean network stabilizes. The nodes in red text were inhibited and nodes in black text were activated.

## Identifying stable motifs

A network attractor is a set of gene states the system can stabilize in *Rozum et al. (2021b)*. The dauer Boolean network had 10 different attractor states (Fig. S1) with one resulting in developmental progression to adult, eight resulting in the dauer fate and one resulting in either dauer or no dauer depending on system conditions (*Rozum et al., 2021a*). Although an attractor represents a potential stable point, not all attractors are accessible from all states. To further analyze system stability, we divided the total network into small subgraphs (*Zanudo & Albert, 2015*; *Zanudo & Albert, 2013*). We identified three Stable Motifs (SMs) in the developmental progression to adult (Figs. 1B–1D). Each of these encompassed genes from at least two different pathways. Stable Motif 1 (SM1; Fig. 1B) spanned 10 genes across the IGF-1/insulin signaling, TGF-β and Steroid hormone pathways (*Hu, 2005-2018*). SM2 and SM3 involved 4 and 8 genes, respectively, across the IGF-1/insulin signaling and TGF-β pathways. *daf-2*, *daf-7* and *hsf-1* (heat shock protein) were common between all three stable motifs with *hsf-1* acting as a crucial node where different pathways intersect. This result has been shown by previous experimental work (*Barna et al., 2012*). Each of the SMs involved a different insulin signaling gene, *ins-1*,

*ins-7* or *ins-18*. We did not identify any SMs for dauer formation (the A3 attractor; Fig. S1).

### Identifying driver nodes in the SMs

We simulated gene perturbations to identify the driver nodes for each of our identified SMs (*Rozum et al., 2021a*). Changing the existing state of the critical nodes shared between the stable motifs (*i.e., daf-2* or *daf-7* downregulation or *hsf-1* upregulation) prevented the system from entering into any of the SMs and pushed it to one of the dauer attractors. Perturbation analysis identified the critical causal interactions in the network as SM2, encompassing *daf-2*, *daf-7*, *ins-7* and *hsf-1*. Perturbation of *daf-2*, *daf-7* or *hsf-1* genes prohibited the system from stabilizing in the SM and the attractors corresponded to each of these nodes. We concluded that *daf-2*, *daf-7* and *hsf-1* were the driver nodes for the SMs. When we perturbed the other nodes of these three stable motifs, the system converged onto either dauer or no dauer *via* different stable motifs. (Table S3). But in case of *daf-2*, *daf-7* and *hsf-1* perturbations, the system did not enter into any of the stable motifs, instead settling into the attractors of those corresponding nodes.

Perturbations of *daf-2*, *daf-7* and *hsf-1* resembled the Daf-c mutant system. When the model was started with dauer-favorable conditions, the system did not have any stable motifs. The same lack of stable motifs occurred with Daf-c mutant phenotypes. Under both dauer-favorable and Daf-c mutant conditions (here, related gene perturbation and/or favorable initial conditions), the system did not have any cyclic subgraphs facilitating control. Instead, the driver nodes decided the fate of the organism in expressing a wildtype or a Daf-c mutant phenotype.

### Analyzing causal interactions in the dauer Boolean network

We identified a reduced network representing causal information flow (*Maheshwari & Albert, 2017*) from environmental input to dauer fate (Fig. 2A). The reduced network contained a cyclic subgraph (Fig. 2B) similar to SM2, containing *daf-2*, *daf-7*, *ins-7* and *hsf-1*. However, it included a necessary inhibitory regulation of *daf-9* by *daf-2*. The dynamics of the entire system can be summarized as input to this subgraph. The full reduced network (Fig. 2C) included 12 nodes and 13 edges.

### Insulin signaling drives the binary dauer/no dauer fate

Gene perturbations revealed an additional seven SMs (SM4-10; Table S3). For each of the perturbations performed, the system could end up in one of 10 different attractors but few of these could be reached from stable conditions with active stable motifs. Activating 'dauer' conditions including low food and pheromone presence resulted in dauer fate and the A3 attractor (Fig. 3A) and available food with pheromone absence resulted in the system reaching the A7 attractor through SMs 1, 2 or 3 (Fig. 3B).

We identified insulin-dependent dauer arrest attractors across the perturbations. Activation of *ins-1* and *ins-18* resulted in the dauer fate through SMs 4-10 but inactivation of either pushed the system to normal development (Fig. 3C). Together with *ins-7* the activation/inactivation profile of these genes pushed the system to normal development through SMs 1, 2 or 3 (Figs. 1B–1D). Similarly, *daf-12* activation resulted in dauer arrest

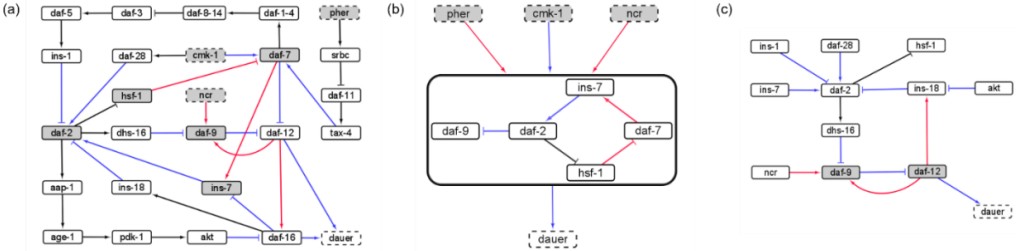

**Figure 2  The causal logic backbone of the dauer Boolean network.** (A) The edges were assigned causal logic (blue = necessary, red = sufficient and black = sufficient and necessary). Nodes filled grey with dashed borders were input signals where grey filled nodes were part of cyclic subgraphs (SMs) and solid borders indicated output nodes. (B) The logic backbone derived from identifying subgraphs from input to motif nodes and motif nodes to output signals. Activation of *pher* and *ncr* was sufficient to reach the motif and *cmk-1* was necessary. The motif stabilization was sufficient to control dauer fate. (C) The reduced dauer logic network consisted of 12 nodes and 13 edges. *Daf-9* and *daf-12* entered a cyclic feedback loop. The insulin factors involved in different endocrine pathways were important points of the network regulating *daf-2* activity and present in the reduced network.

with the insulin profile determining the SM the system entered. When the antagonistic insulins (*ins-1* and *ins-18*) were activated the system stabilized into SM 8 and when *ins-1* was turned off the system stabilized in either SM 2 or 3 (Fig. 3D).

We found that the system must travel through SMs 1, 2 or 3 to end up in normal developmental progression and avoid the dauer fate. To enter dauer arrest the system must travel through one of SMs 4-10 with *ins-1* and *ins-18* activated and *ins-7* inactivated. In this model insulin signaling controls the transitions between normal development and dauer in the presence of gene perturbations. The insulin genes act as decision makers for the convergence of these dauer pathways.

### Identifying binary states across dauer pathways

For normal development we identified one strongly correlated module, one strongly anticorrelated module and a third set of genes with quantitative relationships across the SMs (Fig. 4). For dauer arrest we identified two strongly correlated modules and two strongly anticorrelated modules. These modules span the four signaling pathways involved in dauer arrest and indicate co-regulation and correlative relationships between distant genes in the network.

## DISCUSSION

We have constructed a Boolean model of the dauer arrest network and identified the critical, causal interactions driving the system between dauer fate and normal development. Although the network contains 26 genes and 34 interactions, the system can be driven between the dauer/no dauer fate by controlling a few critical nodes. Importantly, the switchpoints for this system require specific combinations of gene states behaving synergistically or antagonistically. The results of our model demonstrate that complex systems can be controlled through a few critical nodes and connections, echoing both control theory (*Zanudo & Albert, 2015*; *Zanudo, Yang & Albert, 2017*) and long-standing

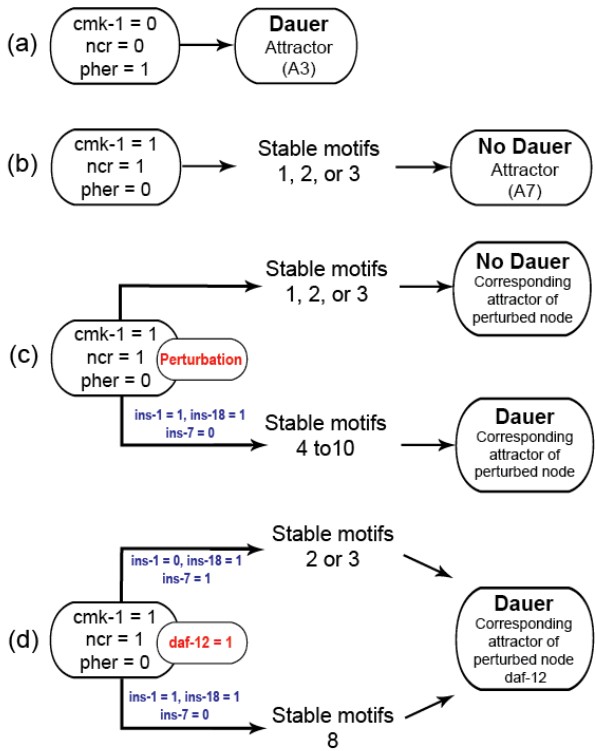

**Figure 3  Model transitions from initial conditions to attractors.** (A) Under unfavourable conditions, the system eventually reached the "Dauer" attractor A3 without settling in any of the cyclic motifs. (B) Under favorable conditions (*cmk-1* is on and pheromone is off), the network stabilized in either of the three cyclic SMs which led to the attractor A7 and the "No dauer" phenotype. (C) When a node of SM 1, 2 or 3 was perturbed the system stabilized in different SMs and converged on either dauer arrest or normal development. (D) *daf-12* perturbation resulted in dauer arrest through SMs 2, 3 or 8. *daf-12* presence, irrespective of initial conditions, was sufficient for dauer formation. SMs 4-10 and the attractors of the perturbed node simulations are given in the Supplementary Materials.

theoretical work on biological systems (*Riedl, 1977*). Evolutionary studies of signaling pathway divergence and conservation have focused on individual genes and sequence-level changes (*Pires-daSilva & Sommer, 2003*). Our results demonstrate that diffuse, correlative interactions have no impact on system stability or developmental fate. We suggest that system-level function and dynamics are, instead, the true evolutionary targets of selection.

Many of the results of our study mirror experimental dauer investigations. For example, a systematic analysis of mutants across insulin signaling genes in *C. elegans* identified roles in lifespan, reproduction, pathogen resistance, thermotolerance and dauer entry and exit (*Fernandes de Abreu et al., 2014*). Here, by constructing a network that encompassed multiple signaling pathways we were able to analyze the role of these insulin signaling genes and robustly identify the interaction between insulin signaling, heat shock response and daf-sensitive genes that determines dauer fate. Our results demonstrate that dauer signaling pathways find different routes to transduce the signals from the environment to developmental decision.
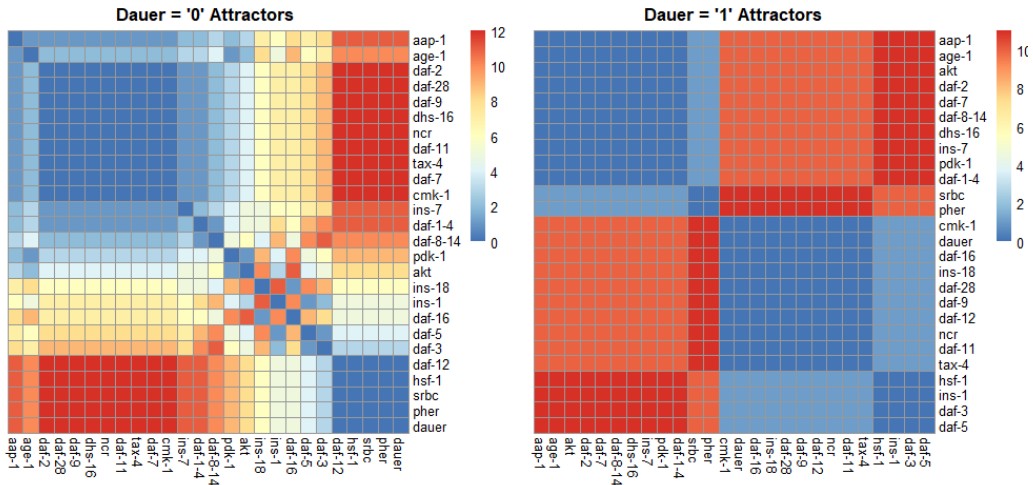

**Figure 4** **Correlation plots as a function of hamming distance for every node of the attractors converging to either "dauer" or "no dauer" when the nodes of the SMs were perturbed.** Hamming distance = 0 implied that the two nodes were in the same state and as the distance value increased the states in which these two nodes exist in the attractors varied. From the dauer =1 correlation plot we observed clusters of nodes that existed in same state across the four dauer inducing pathways. *Ins-1* and *ins-18* were positively and negatively correlated when attractors converged on dauer and no dauer phenotypes, respectively.

Over 100 genes have been described as influencing the dauer fate in *C. elegans* (*Hu, 2005-2018*; *Gilabert et al., 2016*). However, many of these have been identified through mutant analyses or total loss of function alleles (*Fay, 2006*). Genes may be capable of influencing the dauer state through loss-of-function or extreme phenotypes but this may not translate to a gene's involvment in dauer fate in natural populations. We focused on the genes and interactions determined essential to the dauer network. Distilling these further into causal interactions revealed a core set of genes and proteins controlling dauer fate. Transcriptomic studies have identified large, co-regulated, correlated modules associated with dauer mutants, entry and exit (*Fernandes de Abreu et al., 2014*). Our results show that these co-regulated modules also occur in our model (Fig. 4) but their expression and function are correlated consequences of developmental decision-making occurring at a few critical points in the system.

*C. elegans* is a microscopic soil-dwelling organism and extremely dependent on environmental conditions. Some type of developmental arrest or metabolic stasis occurs in many small organisms with similar environmental dependence. For example, tardigrades (Phylum Tardigrada) are microscopic invertebrates closely related to nematodes and can enter both ametabolic and anhydrous states where they extrude all water and remain in suspended animation for multiple years (*Keilin, 1959*). Despite the ubiquity of developmental and metabolic stasis, there is little understanding of the genetics and regulatory systems underlying these processes (*Kalimari et al., 2019*). Evolutionary studies of developmental arrest across nematodes have used dauer genes and pathways as a tool to construct related models. For example, the parasitic nematode *Haemonchus contortus* transitions from both its L3 and L4 stages depending on cues from the host's

gut environment (*Gibbs, 1986*). A dauer model based on *C. elegans* identified 61 gene homologues with varying levels of conservation, each of which may be a potential target for life cycle disruption and anti-parasitic intervention (*Ma et al., 2019*). Alternatively, many of these genes may not be involved in developmental arrest in *H. contortus*. They may have pleiotropic functions in *C. elegans* or have been co-opted for other organismal phenotypes.

Multiple studies support this scenario of conserved dauer phenotype and divergent gene and sequence evolution. A study of dauer-related genes and pathways within Phylum Nematoda found a high number of *Caenorhabditis* and *C. elegans*-specific dauer genes and an evolutionarily recent co-option of TGF-$\beta$ signaling into the dauer pathway (*Gilabert et al., 2016*). Although the insulin signaling pathway is broadly conserved, outside of *Caenorhabditis* the insulin genes examined showed little evolutionary conservation. In the nematode *Pristionchus pacificus* dauer pheromones produce dauer entry in individuals of other genotypes, suggesting the worms are using dauer signaling in intraspecific competition (*Mayer & Sommer, 2011*). Natural variation in *P. pacificus* dauer is partially explained by copy number variation in *dauerless*, an 'orphan' gene that lacks homologs in related species (*Mayer et al., 2015*). These examples of near-scale divergence suggest that although dauer itself is a conserved developmental fate, the individual genes underlying dauer may not be essential targets of selection.

Our results suggest that system-level properties responsible for developmental fate may be the true evolutionary phenotypes under selection. The small number of genes and interactions underlying stable motifs and the causal logic backbones may be subject to higher levels of evolutionary conservation, or their system 'role' may be conserved. Laboratory genetic manipulations within *C. elegans* and across nematodes, focusing on the key genes and interactions identified here, would be a tractable, exciting avenue for future study. For example, within *C. elegans* there are at least 40 insulin signaling homologs (*Girard et al., 2007*). Divergence, loss and duplication of individual genes may not be as significant as the role the homologs together play in dauer signaling.

This work has important future applications. The model can be used to study how different cellular responses work in cohort with dauer pathways by extending the system to encompass genes and interactions involved in disparate pathways including the cell cycle and immune response. Expanding this framework can provide a tractable system to test experimental hypotheses that are currently challenging, labor-intensive and beyond the scope of traditional laboratory investigations. Viewing the conservation and divergence of signaling pathways through the lens of causality and system-level function can provide a robust framework for understanding the evolution of biological networks.

### Funding
JLF and AAK were supported by National Science Foundation grants EF-1921562 and DEB1941854 to JLF. The funders had no role in study design, data collection and analysis, decision to publish, or preparation of the manuscript.

### Grant Disclosures

The following grant information was disclosed by the authors:
National Science Foundation: EF-1921562, DEB1941854.

### Competing Interests

The authors declare there are no competing interests.

### Author Contributions

- Alekhya Kandoor conceived and designed the experiments, performed the experiments, analyzed the data, prepared figures and/or tables, authored or reviewed drafts of the article, and approved the final draft.
- Janna Fierst conceived and designed the experiments, authored or reviewed drafts of the article, and approved the final draft.

### Data Availability

The code, simulations and scripts are available at Github: https://github.com/alekhyaa2/Dauer-Boolean-Model.

The code is available at Zenodo: Kandoor, & Fierst. (2022). Dauer fate in a Caenorhabditis elegans Boolean network model. https://doi.org/10.5281/zenodo.7458189.

### Supplemental Information

Supplemental information for this article can be found online at http://dx.doi.org/10.7717/peerj.14713#supplemental-information.

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
