# Peer review of "Dauer fate in a Caenorhabditis elegans Boolean network model"

_PeerJ, doi:10.7717/peerj.14713_

## Round 0.1 · original submission · Major Revisions

All the reviewers suggested revisions before accepting the manuscript. Kindly address the issues.

Reviewer 1 has suggested that you cite specific references. You are welcome to add it/them if you believe they are relevant. However, you are not required to include these citations, and if you do not include them, this will not influence my decision.

Reviewer 1 ·

Basic reporting

The manuscript is well written and clear. There are some minor points where elements could be explained in more detail or reworded to improve clarity for non-specialists, these are detailed below in the general comments.

Overall the work is described clearly and placed well in its context. The Discussion is however very brief (see comments below).

The literature cited is well referenced and relevant. The overall structure is fine and the figures are clear and well presented. There are however a few points where the some additional explanation, possibly with some additional citations, would improve the detail and accessibility of the work.

Specifically, it might be nice to give a bit more detail about C. elegans when it is introduced. Perhaps this could be done by extending the paragraph at L38-41? Also, at L47-49, the role of pheromone and food are noted in terms of dauer larvae formation, but temperature isn’t mentioned here. It should also be noted that dauer formation in response to just low food or to just high pheromone is not normally seen (e.g. see supplemental data in Jeong et al 2005). You would certainly need to be outside the conditions that would be realistically seen in the wild to induce dauer development in this way. This probably just needs acknowledgement that the switch is based on both inputs.

Discussion of the variation in dauer larvae formation does not refer to earlier work identifying variation between isolates of C. elegans in response to differences in food and pheromone (e.g. Viney et al 2003 and Harvey et al 2008). It would also be useful to note that dauer pheromone is a very complex mix of different chemicals and that their production varies depending on both the environment and the worm (e.g. Butcher et al 2007 and extensive later work from the Butcher and Schroeder labs).

The Discussion is very brief and, with only four citations, doesn’t fully explore the results generated. For example, I wonder if there are there any specific implications for our understanding of dauer larvae development in C. elegans that arise from these findings? It would also be nice to consider if the core elements of the pathway identified here are more broadly conserved across the nematoda than the other genes? Given the earlier assertion about the importance of the approach, it would also be nice to see the final paragraph (L207-211) extended.

Butcher, R.A., Fujita, M., Schroeder, F.C. and Clardy, J., 2007. Small-molecule pheromones that control dauer development in Caenorhabditis elegans. Nature chemical biology, 3(7), pp.420-422.
Harvey, S.C., Shorto, A. and Viney, M.E., 2008. Quantitative genetic analysis of life-history traits of Caenorhabditis elegans in stressful environments. BMC evolutionary biology, 8(1), pp.1-16.
Viney, M.E., Gardner, M.P. and Jackson, J.A., 2003. Variation in Caenorhabditis elegans dauer larva formation. Development, growth & differentiation, 45(4), pp.389-396.

Experimental design

The manuscript reports original work that is within the scope of PeerJ. The question addressed is well defined, relevant and meaningful. I found the Methods to be clear and, given the links to the code etc, it is my opinion that the analysis could be replicated.

Validity of the findings

Raw data is supplied and is clearly annotated. It might however be nice to have included the original longer list of dauer-related genes.

Additional comments

L21-23 the two parts of this sentence don’t seem to read clearly to me, could this be reworded to make it clearer what is interacting with daf-12?
L117-119 Sentence here seems unclear. Can this be reworded?
Throughout there are multiple uses of “but” that do not have commas before them. The commas should be added in these cases, and other points in the manuscript where sub-clauses aren’t separated should also be checked.

·

Basic reporting

In this study, the authors constructed a boolean model to decipher the gene relationships among different, physically separated signaling pathways in order to understand the C.elegans dauer/no dauer state progression.

Through computational network simulation and perturbation, authors identify 3 stable motifs in the developmental networks that are enriched in genes involving in insulin profiling. Authors surmised that insulin profile determines the stable motifs and daf-12 serves as a switch point for dauer/no dauer transition.

In my opinion, this is an interesting work that helps to distill complex biological regulation into simpler decision flows. In addition to explaining C.elegans dauer state, the methodology described here can be applicable to other biological systems as well.

Overall, the manuscript uses a coherent language, references are appropriately cited, and sufficient background is provided. The associate computer code makes sense to me.

However, the study requires major revision on certain fronts before it can be accepted for publication. The quality of figures is generally not of premium nature. The methods section is not described in sufficient detail. Additionally, in my opinion an experimental validation of the conclusion of this computational work should be carried out so as to validate the findings. Alternatively, suitable proof (from literature or computationally) should be provided. Please see my additional comments for suggestions and details.

Experimental design

Overall, methods are not described in sufficient detail. The conclusion can also talk about how this kind of boolean model based network analysis can have wider appeal for distilling complex biological processes into simpler explanations. Authors mention this a bit in Lines 209-211. However, it could be beneficial to expand on this point in the conclusion.

Validity of the findings

The raw data provided is satisfactory. methods are missing details about the parameters and statistics used. Please see my additional comments for details.

Additional comments

Additional Comments:

1. Lines 50-52: Authors mention that entry and exit into dauer state occurs at variable rates. However, the model described in this article is Boolean with 2 outcomes – ON or OFF. How can the observed variable rates of dauer phase initiation and exit be explained using the model described in this work? Are there any sensitivity parameters or cutoffs associated with the model build?
2. Line 69: How many total genes were extracted? Please mention exact number. If there was any filtering involved, mention that as well.
3. Lines 70-72: More information is needed with regards to identification of neighboring genes. (i) How many genes were identified?; (ii) How many were 1o neighbors of shortlisted genes; (iii) Were the identified genes physical or functional interactors of dauer genes?; (iv) What parameters were employed to identify these genes (e.g. ; neighborhood connectivity, nearest neighbors etc.)?
4. Lines 73-75 and Supplementary table 1: How was the quality of scientific articles utilized to obtain gene interactions verified? I assume not every interaction will come from equally high-quality scientific article. Additionally, when multiple articles describe a given interaction, how is this information considered? The methodology or flow for this should be described in methods section.
5. Lines 77-86: Since development of Boolean network forms the basis of this work, it needs to be described in more detail. A separate section should be created in the methods describing different steps.
6. Line 95: What is the meaning of the term “asynchronous updating” in terms of simulation parameters? Are there any other comparable updating methods that this process was compared with? Authors mention a brief description in line 96-97, however, explanation with respect to their experiment is needed to showcase the benefits of this method.
7. Lines 97-100: What was the rationale for performing 100 simulations? Authors mention that the nodes reached equilibrium after 15 iterations. This information should be shown as separate plots depicting all iterations and clearly showing the iteration# where equilibrium has been reached. Authors could use network parameters to show this information.
8. Lines 110-114: Authors mention that KO Daf-c genes separately forms dauer. However, supplementary figure 1b appears to show that all Daf-c genes (daf-2, daf-7, daf-11, daf-11) were KO together? Same appears to be true for daf-d mutants.
9. Lines 116-120: The terms attractors, motifs and perturbations need to be defined in more detail in the methods with examples from this study. This can aid in increasing the readership of this work. Also, the settings used in the pystablemotifs tool should be mentioned.
10. Lines 145-151: Authors changed the existing states of the critical nodes to identify driver nodes. I imagine changing the existing states of non-critical nodes (not part of stable motifs in Figure 1b-d) could also lead to an impact on the system output? For clarity, authors can run some control experiments where they change the existing states of non-critical nodes and show the outcomes.
11. Lines 168-175: Through networks simulation and perturbation, authors show that insulin genes are drivers for dauer pathways. If true, this information can be significant in understanding this process. A wet lab experiment should be conducted to validate this finding where selected insulin genes identified here can be activated or repressed.
12. Lines 193-194: Why does impact of loss-of-function of genes on dauer state does not translate to gene’s involvement in the dauer/no dauer fate?

Minor

1. Methods section does not follow a comprehensible narrative. Methods need to be described in individual sub-sections describing the steps, parameters, settings.
2. In my opinion, all the supplementary figures and tables should be part of the main manuscript. They seem critical to understand the described results.
3. Cytoscape networks are generally of low quality. Font sizes are almost un-readable and the networks are lacking a legend for edges.

Reviewer 3 ·

Basic reporting

The text in figures needs to be larger and the graphic needs to be higher quality. Arrow heads and “T” edges need to be more pronounced in figure 1.
A small type in Line 207, “The model can used” should be “The model can be used”.

Experimental design

It would be better if the authors could provide more details on why they chose the “asynchronous updating” scheme.

Validity of the findings

The authors surveyed the C. elegans dauer formation pathways in the literature exhaustively. They extracted from empirical data and constructed a solid network to start with for Boolean network modeling. They successfully identified multiple stable motifs and driver nodes in each SM through perturbation analysis.
The main limitation is that the bioinformatics conclusions were not validated experimentally. Although the authors commented on the validity of their predictions in the discussion section, the significance of the bioinformatic findings is still somewhat lacking. I understand that constructing strains containing multiple mutations is cumbersome and maybe not in the scope of this study, it would still be exciting if the authors could give some pointers toward future directions experimentally, especially for less complex SMs, such as SM2.

---

## Round 0.2 · accepted · Accept

Both the reviewers are in favor of the manuscript hence my recommendation is to accept

Reviewer 1 ·

Basic reporting

Writing is clear and appropriate.

Experimental design

The manuscript reports original work and the methods are clear.

Validity of the findings

Results and conclusions clear. Changes to the Discussion place the results in context.

Additional comments

Thank you for addressing my comments on the original version of this manuscript.

·

Basic reporting

.

Experimental design

.

Validity of the findings

.

Additional comments

Authors have responded to majority of my comments and I have no pending issues. I commend the authors for adding the Supplementary Figure 2. This figure adds to the understanding of this work.

I am not completely convinced with the conclusion that insulin genes are most important based on this work. Without experimental validation, this point cannot be made in my opinion. However, authors have satisfactorily explained the limitations of performing wet lab work from their end and hence I leave this to the editor to make a final decision.

All in all, this work could serve as an interesting starting point for performing this kind of network analyses for multiple biological processes.